# Effect of CO Molecule Orientation on the Reduction of Cu-Based Nanoparticles

**DOI:** 10.3390/nano11020279

**Published:** 2021-01-22

**Authors:** Sergey Y. Sarvadii, Andrey K. Gatin, Vasiliy A. Kharitonov, Nadezhda V. Dokhlikova, Sergey A. Ozerin, Maxim V. Grishin, Boris R. Shub

**Affiliations:** N.N. Semenov Federal Research Center for Chemical Physics, Russian Academy of Sciences (FRCCP RAS), Kosygina str. 4, 119991 Moscow, Russia; akgatin@yandex.ru (A.K.G.); vch.ost@mail.ru (V.A.K.); dohlikovanv@gmail.com (N.V.D.); sergeoz@yandex.ru (S.A.O.); mvgrishin68@yandex.ru (M.V.G.); bshub@mail.ru (B.R.S.)

**Keywords:** copper oxides, CO adsorption, reduction, electric field, scanning tunneling microscopy, scanning tunneling spectroscopy

## Abstract

The adsorption of CO on the surface of Cu-based nanoparticles was studied in the presence of an external electric field by means of scanning tunneling microscopy (STM) and spectroscopy (STS). Nanoparticles were synthesized on the surface of a graphite support by the impregnation–precipitation method. The chemical composition of the surface of the nanoparticles was determined as a mixture of Cu_2_O, Cu_4_O_3_ and CuO oxides. CO was adsorbed from the gas phase onto the surface of the nanoparticles. During the adsorption process, the potential differences Δ*V* = +1 or −1 V were applied to the vacuum gap between the sample and the grounded tip. Thus, the system of the STM tip and sample surface formed an asymmetric capacitor, inside which an inhomogeneous electric field existed. The CO adsorption process is accompanied by the partial reduction of nanoparticles. Due to the orientation of the CO molecule in the electric field, the reduction was weak in the case of a positive potential difference, while in the case of a negative potential difference, the reduction rate increased significantly. The ability to control the adsorption process of CO by means of an external electric field was demonstrated. The size of the nanoparticle was shown to be the key factor affecting the adsorption process, and particularly, the strength of the local electric field close to the nanoparticle surface.

## 1. Introduction

Copper nanoparticles are widely used in modern catalysis. They are particularly attractive because of copper’s high natural abundance, low cost and the multiple practical and straightforward ways of preparing Cu-based nanomaterials. They are used in the azide–alkyne cycloaddition process [1,2,3] and in the synthesis of 1,2,3-triazoles [4]. Due to their low cost, copper nanoparticles have successfully replaced catalysts based on noble and transition metals in Heck and Sonogashira reactions, as well as in the reactions of C–S, C–O, C–B, and C–Se bonds formation [5,6,7,8,9]. In [10,11], copper was used as the main component of polymetallic catalysts. In [12,13], metal–organic and composite systems based on copper were considered. Despite numerous studies on Cu-based catalysts [14], this area of research still has many aspects that remain to be explored.

Due to the charge transfer between nanoparticles and the support, a local electric field of a sufficiently high strength of 10^6^–10^9^ V/m can exist close to the nanoparticle surface. The presence of such a field can affect the rate of chemical processes on the surface of the nanostructured coating. Indeed, the orientation of polar molecules induced by an electric field can lead to the synthesis of different products. The ability to control chemical reactions by electric fields was predicted in theory in [15], where the choice of orientation and direction of the field vis-à-vis the molecular axes was shown to drive the reaction in the direction of complete C−H hydroxylation or complete C=C epoxidation. This has been experimentally proven recently by an example of the Diels–Alder reaction, which was studied by the scanning tunneling microscopy (STM) break-junction method [16]. Previously, we found similar effects of the electric fields on the reduction of oxidized platinum by hydrogen [17], hydrogen adsorption on gold [18,19], decomposition of ammonia on organoboron [20] and platinum nanoparticles [21], and some reactions of the hydrogenation of hydrocarbons [22].

The methods and conditions used for the synthesis of nanoparticles mainly determine the possibilities of their further use. Therefore, it is important to accurately determine the characteristics of synthesized nanoparticles: their shape, size, and chemical composition [23,24]. Typically, catalysts are studied by standard methods, even in the case of nanostructured ones. Such methods provide information averaged over a large ensemble of nanoparticles, thus this approach practically does not allow the detection of differences between individual nanoparticles. To solve this problem, new approaches are required with local chemical sensitivity and high spatial and temporal resolution. These requirements are met by scanning probe methods, particularly by scanning tunneling microscopy and spectroscopy (STM/STS) methods [25,26,27,28,29]. By these methods, one can obtain the most detailed information on the structure and properties of nanostructured systems.

The aim of our work is to study the effect of CO orientation in an external electric field on the reduction of Cu-based nanoparticles and the distribution of various oxide phases on the nanoparticle surface.

## 2. Materials and Methods

Nanoparticles were synthesized on the surface of highly oriented pyrolytic graphite (HOPG) with an angular spread of the *c*-axes of the crystallites of 0.8° [30] by the impregnation–precipitation method. An aqueous solution of copper(II) nitrate Cu(NO_3_)_2_ with a concentration of 24 mg/L was used as a precursor. The precursor solution was applied to a cleaned surface of HOPG (AIST-NT, Moscow, Russia), which looked like vast atomically smooth C(0001) terraces. After drying the solution, the sample was placed in an STM chamber, where it was calcined under ultrahigh vacuum (UHV) conditions at a temperature of 750–850 K for 8 h. The residual gas pressure did not exceed 2 × 10^–10^ mbar. During the calcination, the precursor solution decomposed with the formation of copper(II) oxide and gaseous nitrogen(IV) oxide and oxygen, which were pumped out of the chamber. The decomposition of anhydrous copper(II) nitrate at atmospheric pressure is known to begin at a temperature of 440 K [31]. The precursor decomposition occurs:(1)$$2Cu{\left(NO_{3}\right)}_{2}\overset{t}{\to }2CuO+4NO_{2}+O_{2}$$

The experiments were carried out with a scanning tunneling microscope (UHV VT STM, Omicron NanoTechnology, Taunusstein, Germany) under ultrahigh vacuum conditions. This allowed us to avoid uncontrolled changes in the chemical composition of the samples due to residual gases. The platinum–iridium STM tips used for the experiments were prepared by standard methods and treated by argon ion sputtering under UHV conditions. In the experiments, we used only those tips that showed reproducible volt-ampere characteristics (VAC) with S-shaped curves when scanning HOPG. Such VACs are typical for metal–metal tunnel nanocontacts [25,26,27,28,29].

The CO adsorption under 1 × 10^−6^ mbar of pressure and the reduction of nanoparticles were studied at 300 K in the presence of an external oriented electric field. Before introducing the gas, the STM tip was removed ~5 mm away from the surface of the sample, and during the adsorption experiment, the tip was fixed in this position. During exposure, a potential difference, Δ*V*, was applied to the vacuum gap between the sample and the grounded tip. Thus, the system of the STM tip and sample surface forms an asymmetric capacitor, inside which an inhomogeneous electric field exists. Exposure of the samples to CO was carried out at two potential differences: Δ*V* = +1 and −1 V with respect to the grounded STM tip. After pumping CO out of the chamber, the potential difference was switched off, and STM spectroscopy experiments were performed to investigate the structure of Cu-based nanoparticles modified by CO. The exposure value was measured in Langmuir, 1 L = 1.33 × 10^−6^ mbar·s.

## 3. Results and Discussion

### 3.1. Morphology, Local Electronic Structure and Chemical Composition

The shape of copper nanoparticles in most cases was found to be close to spherical, and their diameter was about 10 nm (Figure 1). Most of the nanoparticles are agglomerated and form irregularly shaped clusters consisting of hundreds of nanoparticles. They are grouped near the boundaries of the graphene sheets that make up the surface of the support. About 20% of the sample surface is covered with nanoparticles.

STM measurements have shown that the surface of the nanoparticles is rather inhomogeneous. The local electronic structure changes significantly even within the surface of a single nanoparticle. In general, we can say that the electronic structure of the nanoparticles corresponds to a semiconductor; that is, the nanoparticles consist of copper oxides, but in rare cases, there are areas of pure metal on the surface. Local nanocontact VACs, measured at different points of the nanoparticle surface, can be conventionally divided into four types. Figure 2 shows various types of VAC behavior. About 23% of VAC curves correspond to Curve 1. In these curves, the zero-current section ranges from 0 to 1.1 V. This also includes surface areas with a metallic type of conductivity. Hereafter, the local electronic structure of this type will be denoted as Cu-1.

Local VAC curves similar to Curve 2 in Figure 2 occur in 30% of cases. For such curves, the zero-current section is about 1.2–1.8 V. The electronic structure of the synthesized nanoparticles at these points on the surface corresponds to a semiconductor. This suggests that oxide is present at least on the surface of the nanoparticles. This type of electronic structure will be denoted as Cu-2.

The VAC curves (Figure 2, Curve 3) with a section of zero-current of 2.2–2.8 V are found in 30% of cases. The surface of the nanoparticles at these points is also covered with oxide, but its characteristics differ from the previous ones. We will denote such a type of electronic structure as Cu-3.

Finally, the VAC curves in 14% of cases demonstrate a zero-current section of 2.9 V and above (Figure 2, Curve 4). We will denote this type of electronic structure as Cu-4. Additionally, we should say that VACs with a section of zero-current of 1.9–2.1 V practically do not occur (2–3%) during all the experiments; therefore, hereafter, we will only talk about the four types of electronic structure listed above.

Since the width of the zero-current section corresponds to the band gap of the material under the STM tip up to a dimensional factor, one can conclude that the local electronic structure of Cu-based nanoparticles deposited on HOPG belongs to one of four types: Cu-1 with a small band gap (no more than 1.1 eV), Cu-2 with a band gap of about 1.2–1.8 eV, Cu-3 with a band gap of about 2.2–2.8 eV, and Cu-4 with a band gap of more than 2.9 eV. In Figure 2, the band gap edges for various VAC types are marked with colored arrows. The initial distribution of various types of local electronic structure over the surface of synthesized nanoparticles is shown in Figure 3a.

The results obtained can be compared with known data and the local chemical composition of various surface areas can be analyzed. The electronic structure of CuO, Cu_2_O and Cu_4_O_3_ copper oxides is considered in a number of works [32,33,34,35,36,37]. According to them, all these oxides are semiconductors, and Cu_2_O has the largest band gap of 2.17–2.62 eV [33], and the band gaps of CuO and Cu_4_O_3_ are in the range of 1.35–1.7 eV [33,34,35,36,37]. One can conclude that the electronic structure of the type Cu-2 corresponds to a mixture of CuO and Cu_4_O_3_ oxides, the electronic characteristics of which are similar. The local chemical composition of the surface areas with an electronic structure of the type Cu-3 corresponds to the Cu_2_O oxide. This conclusion is also confirmed by the fact that during all the experiments, practically no surface areas with a local band gap of 1.9–2.1 eV were found. Thus, a mixture of CuO and Cu_4_O_3_ oxides always differs from Cu_2_O in spectroscopy.

Oxide structures of the type Cu-1 on the surface of the nanoparticles do not have an exact correspondence with the known stable copper oxides. The known band gap values in this case exceed the observed ones. This situation can be realized in the case of the formation of a very thin oxide layer on the metal or a low content of oxygen atoms in the oxide (<0.5 at.O/at.Cu). In the case of Cu-4, apparently, the observed band gap is associated not with the chemical composition of the formed oxide, but with its high defectiveness. One should remember that the electronic structure of the nanoparticles can be strongly distorted due to interaction with substrate defects, the electronic structure of which is also very different from the electronic structure of a defect-free graphite surface [38].

### 3.2. CO Adsorption Experiments in Presence of External Electric Field

Since the CO molecule is polar, one can assume that the electric field inside the asymmetric capacitor formed by the STM tip and the sample surface can affect the orientation of the CO molecule. Therefore, by changing the direction of the electric field, it is possible to prevent or promote the reduction of copper on the surface of nanoparticles. In this case, due to the small radius of curvature (*ρ* ~ 10^−9^ m), the strength of the local electric field, *E*, near the surface of nanoparticles can be very high (*ρE* = *const*). To test this assumption, we performed a CO adsorption experiment, during which the potential difference, Δ*V* = +1, or −1 V, was applied to the vacuum gap between the grounded tip and the sample.

The exposure of the sample to CO (400 L) at a potential Δ*V* = −1 V does not influence the size of the nanoparticles or their spatial distribution, but significantly changes the electronic structure of the nanoparticle surface. One can see that the ratio between various types of local electronic structure changes dramatically (Figure 3b): Cu-1 increases to 54%; and other types—Cu-2, Cu-3 and Cu-4—decrease to 20%, 17% and 7%, respectively. At the same time, among the VAC curves, a large proportion without a zero-current section is found (included in Cu-1). That is, a partial reduction of copper occurs on the surface of nanoparticles. This is consistent with the results of [39], where it was noted that the complete reduction of oxidized copper in CO can occur even at a relatively low temperature of 473 K.

Exposure of the sample to air (40 h) again significantly changed the electronic structure of the surface (Figure 3c): the content of Cu-1 decreased to 18%; Cu-2 and Cu-3 increased to 41% and 25%, respectively; and Cu-4 practically did not change (5%). It should be noted that the system obtained by oxidation in atmospheric oxygen is somewhat different from the system obtained by precursor thermal decomposition in UHV. Apparently, the reason for this is that during the thermal decomposition of the precursor, the formation of oxide phases occurs in parallel. Since all oxygen and nitrogen oxides are immediately pumped out from the system, these reactions can be considered quasi-independent. However, during oxidation in the atmosphere, oxide phases are formed sequentially under conditions of excessive oxygen content. In this case, the oxides Cu_2_O and Cu_4_O_3_ are intermediate products, and their concentration will strongly depend on the concentrations of all other Cu-based phases:(2)$$Cu\overset{+O_{2}}{\to }Cu_{2}O\overset{+O_{2}}{\to }Cu_{4}O_{3}\overset{+O_{2}}{\to }CuO$$

This assumption is consistent with the results of [40], from which it follows that the Cu_4_O_3_ or Cu_2_O oxides can be both intermediate and independent products of reactions of copper and its oxides with gases.

After STM measurements, the sample was, again, exposed to CO (400 L). During exposure, a potential difference Δ*V* = +1 V was applied to the vacuum gap. At this stage, the electronic structure of the sample surface was rearranged again (Figure 3d): the content of Cu-1 increased to 36%, and Cu-2 and Cu-3 decreased to 27% and 19%, respectively. The content of Cu-4 increased slightly to 15%.

### 3.3. Simulation

Since Cu-1 corresponds to metallic copper and unstable oxides with a low oxygen content (<0.5 at.O/at.Cu), the content of Cu-1 in the system can be taken as a measure of reduction efficiency. Comparing the state of the system after reduction at various potential differences applied to the vacuum gap, one can note that when the potential difference Δ*V* = −1 V is applied, reduction is more efficient: the content of compounds with an electronic structure of the Cu-1 type in this case increases from 23% to 55%, while at the positive potential, from 18% to 36%.

Initially, it was assumed that the orientation of the CO molecule with its carbon end to the nanoparticle surface should facilitate copper reduction. The obtained result is consistent with this assumption. Indeed, at Δ*V* = −1 V, the sample is negatively charged, and the CO molecule is turned towards the nanoparticle surface by a carbon atom, on which there is a partial positive charge δ+.

According to [41], the reduction of copper oxides in CO can vary considerably with experimental conditions (gas flow rate, temperature, sample size, etc.), and its complex kinetics cannot be described by a single n-order expression over the entire range of the reaction. In order to make simple estimations, it is enough to consider this process as a complex reaction proceeding according to the scheme [39]:(3)$$Cu_{ox}+CO\;\begin{array}{c} \overset{k_{1}}{\to }\\ \underset{k_{2}}{\leftarrow } \end{array}\;Cu_{ox}\left(CO\right)\;\overset{k_{3}}{\to }Cu_{red}+CO_{2},$$
where *k*_1_, *k*_2_, and *k*_3_ are reaction rate constants; *Cu_red_* are compounds with electronic structure of type Cu-1; and *Cu_ox_* are compounds with electronic structure of types Cu-2, Cu-3 and Cu-4. The analytical solution of a system of differential equations corresponding to a given reaction scheme allows the concentration of the *Cu_red_* product to be expressed [42]:(4a)$$\Delta C_{red}=C_{ox}^{0}\left[1-\frac{\alpha +\beta }{2\beta }exp\left(-\frac{\alpha -\beta }{2}t\right)+\frac{\alpha -\beta }{2\beta }exp\left(-\frac{\alpha +\beta }{2}t\right)\right],$$
(4b)$$\alpha =k_{1}C_{CO}+k_{2}+k_{3},\;\;\beta =\sqrt{\alpha^{2}-4C_{CO}k_{1}k_{3}},$$
where *C_red_* is the concentration of compounds with electronic structure of type Cu-1, *C_CO_* is the concentration of CO molecules turned with a carbon end to a nanoparticle, and $C_{ox}^{0}$ is the initial concentration of all copper oxide phases except Cu-1 in the system. Expression (4a) describes not the *C_red_* as it is, but the difference between its final and initial values. The reason for such a result is that Expression (4a) is obtained under the assumption that the final stage of the reaction is irreversible, as CO_2_ is being pumped out of the system during the reaction. Thus, the CO_2_ concentration and initial concentration of metal copper would not influence the reaction rate. The *C_CO_* remains constant during the experiment. Work [39] makes it possible to estimate the values of *k*_1_, *k*_2_ and *k*_3_ for processes involving CuO or Cu_2_O at room temperature (Table 1).

For both reactions, *C_CO_* ≈ 5 × 10^–8^ mol/m^3^, *t* ≈ 400 s, and *k*_1_*C_CO_* << *k*_2_, *k*_3_; therefore, we can write the following:(5a)$$\frac{\alpha -\beta }{2\beta }\approx \frac{k_{1}C_{CO}}{k_{2}+k_{3}},\;\;$$
(5b)$$\frac{\alpha +\beta }{2\beta }\approx 1,$$
(5c)$$exp\left(-\frac{\alpha -\beta }{2}t\right)\approx 1-\frac{k_{1}C_{CO}}{2},\;\;$$
(5d)$$\frac{\alpha -\beta }{2\beta }exp\left(-\frac{\alpha +\beta }{2}t\right)\to 0$$

As a result, we obtain the following:(6)$$\Delta C_{red}\sim C_{ox}^{0}C_{CO}k_{1}t.$$

Since the reaction $Cu_{2}O↔CuO+Cu$ practically does not occur [39], the reduction processes of CuO and Cu_2_O can be considered separately; that is, Expression (6) is valid regardless of the chemical composition of the oxide. In this case, the experimental results can be described by a simple two-level model, in which CO molecules can be oriented “along the field” and “against the field”. If the ground state of the CO molecule is considered to be orientated “along the field”, we obtain:(7)$$C_{CO+}=C_{CO-}\times exp\left(-\frac{Ed}{k_{B}T}\right)$$
where *k_B_* is the Boltzmann constant; *E* is the electric field strength close to the surface of the nanoparticle; *d* is the electric dipole moment of the CO molecule; *C_CO–_* and *C_CO+_* are the concentrations of CO molecules oriented “along the field” and “against the field”, respectively. One can assume that in the case of a negative potential Δ*V* = −1 V, reduction occurs only due to the molecules turned by their carbon end to the nanoparticle oriented “along the field”. In the case of a positive potential Δ*V*= +1 V, everything happens the other way: reduction occurs only due to the disoriented molecules. Before exposure to CO, the initial concentration of $C_{ox}^{0}$ in the systems is approximately the same: 0.77 in the case of Δ*V* = –1 V and 0.82 in the case of Δ*V* = +1 V. The concentration here is the relative content of this oxide phase (Figure 3a,c). Therefore, the following is valid:(8)$$\frac{\Delta C_{red+}}{\Delta C_{red-}}\approx \frac{C_{CO+}}{C_{CO-}}=exp\left(-\frac{Ed}{k_{B}T}\right)$$
where Δ*C_red–_* and Δ*C_red+_* are the changes in concentrations of compounds with electronic structure of the type Cu-1, when the potential difference is positive and negative correspondingly. Since *d* is 0.4 × 10^–30^ C·m for CO molecule, *C_red–_* changes from 0.23 to 0.54, and *C_red+_* changes from 0.18 to 0.36 (Figure 3b,d), one can estimate the strength of the electric field required for the orientation effect to be observed:(9)$$E=-\frac{k_{B}T}{d}\ln\left(\frac{\Delta C_{red+}}{\Delta C_{red-}}\right)\approx 6\times 10^{9}V/m.$$

The obtained value is consistent with the assumption of a strong electric field close to the surface of nanoparticles and with the results of works [19,43].

Thus, the orientation of the CO molecule with the carbon end to the nanoparticle promotes the reduction of copper in CuO, Cu_4_O_3_, and Cu_2_O oxide systems. Since the key factor affecting the local electric field strength is the characteristic size of the elements of the nanostructured coating, this effect can be observed in weak electric fields if the nanoparticle size is 1–10 nm.

## 4. Conclusions

The experiments described above have demonstrated the possibility of controlling the adsorption of polar molecules on the surface of nanoparticles by means of an external electric field. STM/STS methods have been successfully used to study the adsorption of CO on Cu-based nanoparticles deposited on the surface of highly oriented pyrolytic graphite. It was shown that the orientation of the CO molecule with the carbon end to the nanoparticle promotes metal reduction. When the potential difference Δ*V* = −1 V is applied to the sample with respect to the grounded STM tip, the exposure of the sample to CO (*T* = 300 K and *P* = 1 × 10^–6^ mbar) is accompanied by CO adsorption and subsequent copper reduction. In the case of Δ*V* = +1 V, CO adsorption and copper reduction efficiency decrease. This result is well described by the two-level model. The performed estimates of the electric field strength confirm that the orientation effect manifests itself close to the surface of nanoparticles with a radius of curvature of ~10^−9^ m.

## Figures and Tables

**Figure 1 nanomaterials-11-00279-f001:**
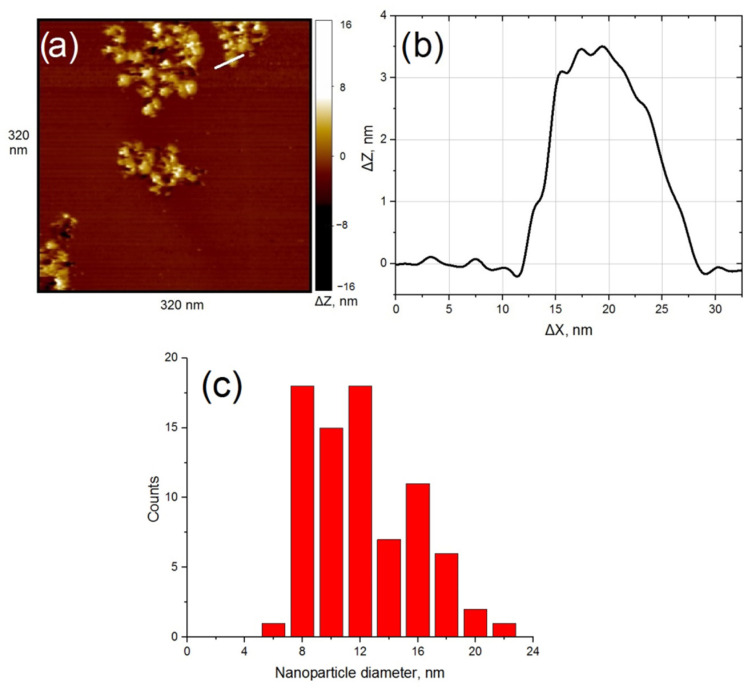
The sample after calcination under ultrahigh vacuum (UHV) conditions (*t* = 8 h, *T* = 750−850 K). Results of the scanning tunneling microscopy (STM) measurement: (**a**) topography image of highly oriented pyrolytic graphite (HOPG) surface with deposited clusters of Cu-based nanoparticles; (**b**) profile of the surface along the cut line shown in Figure 1a; (**c**) nanoparticle size distribution.

**Figure 2 nanomaterials-11-00279-f002:**
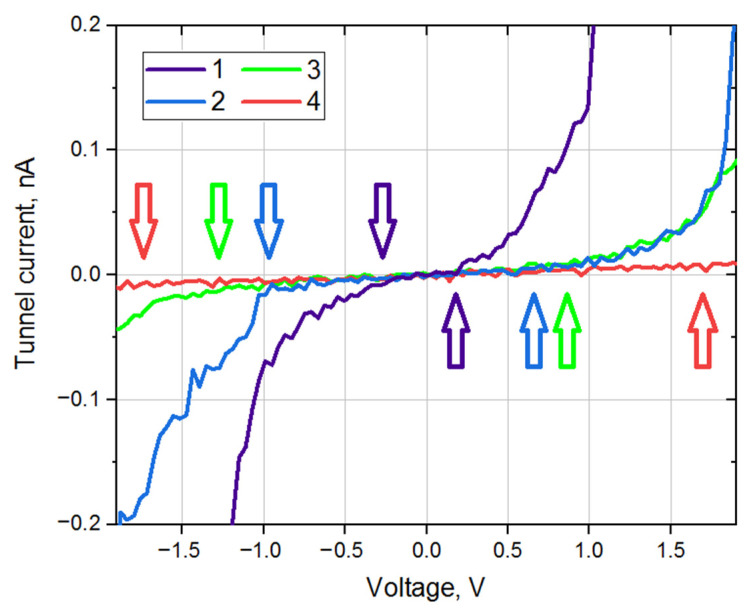
Types of local volt-ampere characteristic (VAC) curves, corresponding to various copper oxides. The edges of zero-current sections for various curves are marked with colored arrows.

**Figure 3 nanomaterials-11-00279-f003:**
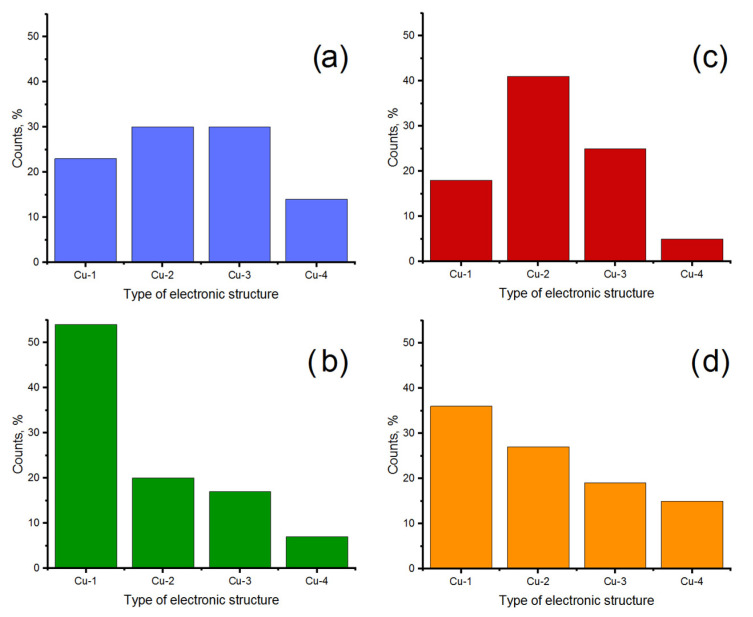
Ratio between various types of local electronic structure on the surface of nanoparticles: (**a**) after calcination and precursor decomposition; (**b**) after exposure to CO (400 L, *T* = 300 K, Δ*V* = −1 V); (**c**) after oxidation in air (*t* = 40 h, *T* = 300 K); (**d**) after exposure to CO (400 L, *T* = 300 K, Δ*V* = +1 V). To obtain each histogram, at least 50 counts were performed.

**Table 1 nanomaterials-11-00279-t001:** Values of *k*_1_, *k*_2_ and *k*_3_ for processes involving CuO or Cu_2_O at room temperature according to [39].

	*k*_1_, m^3^·mol^–1^·s^–1^	*k*_2_, s^–1^	*k*_3_, s^–1^
*CuO → Cu*	10^–5^	10^–1^	1
*Cu* _2_ *O → Cu*	10^–7^	10^–1^	10^–4^

## Data Availability

The data presented in this study are available on request from the corresponding author.

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
