# Peer review of "Effect of CO Molecule Orientation on the Reduction of Cu-Based Nanoparticles"

_nanomaterials, 2021, doi:10.3390/nano11020279_

Round 1

Reviewer 1 Report

The authors researched the adsorption of CO on the surface of Cu-based nanoparticles in the presence of an external electric field and with the means of  STM and STS. They inspected the effect of two potential differences on CO orientation in an external electric field on Cu-based nanoparticles and the distribution of various Cu-oxide phases on the nanoparticle surface. This work presents interesting data and valuable means. However, to be accepted and published, the authors need to give consideration on the following points:

  1. Abstract: The discription of potential differences ΔV = +1 or –1 Ð’ is wrong, please check it.
  2. How about the stability of the material?
  3. The authors wrote:” The ability to control 46 chemical reactions by electric fields was predicted in theory in [14].” Here, the sentence is incomplete.
  4. The research progress on Cu-based catalysts needs to be supplemented slightly, and the reasons for using Cu-based catalysts should be clear.
  5. The diameter of the Cu nanomaterial of 10 nm is not clear from Fig 1, and the description of "20%of the sample surface is covered with nanomaterials" is also not So, it is suggested to provide a high resolution graph for readers.
  6. From Fig 1, it can be seen that the as-synthetized nanomaterial is a little agglomerated, indicating a low and poor dispersity.
  7. The symbol of ÷ is wrong in the formulation like 1.2÷1.8 V, 2.2÷2.8 V, etc.
  8. In the nanomaterial, the CuO, Cu2O and Cu4O3 made up it. I want to know how to control their ratio and content in the nanomaterials.
  9. Why did the authors say "the concentration of Cu2O and Cu4O3 strongly depend on the concentrations of all other reagents"? and what are the other reagents ?
  10. How to prove the process of Cu2O to Cu did not occur?

Author Response

To Reviewer #1

Many thanks for your valuable comments that helped us to improve the quality of the manuscript.

  1. Abstract: The description of potential differences ΔV = +1 or –1 Ð’ is wrong, please check it.

Answer: We agree with this amendment. Relevant changes have been added to the manuscript – line 19.

  1. How about the stability of the material?

Answer: The synthesized material is stable. During the exposure to CO we haven’t observed any changes in the size of the nanoparticles or in their spatial distribution. Relevant comment has been added to the manuscript – lines 193-194.

  1. The authors wrote:” The ability to control 46 chemical reactions by electric fields was predicted in theory in [14].” Here, the sentence is incomplete.

Answer: We agree with this amendment. Relevant changes have been added to the manuscript. The results of the article [15/ old 14] were described more clearly – lines 51-52.

  1. The research progress on Cu-based catalysts needs to be supplemented slightly, and the reasons for using Cu-based catalysts should be clear.

Answer: Thanks a lot for this amendment. Relevant changes have been made to the manuscript – lines 35-37 and 43-45, and ref [14 - DOI: 10.1021/acs.chemrev.5b00482] was added.

  1. The diameter of the Cu nanomaterial of 10 nm is not clear from Fig 1, and the description of "20% of the sample surface is covered with nanomaterials" is also not. So, it is suggested to provide a high resolution graph for readers.

Answer: Thank you for this amendment. We have included the data on the nanoparticle size distribution to the manuscript – Figure 1c. As for “20%”, this value was obtained for big areas of HOPG surface covered with nanoparticles. Due to inhomogeneity of the covering, for smaller areas the local filling degree can be a bit different. We suppose Figure 1a to be enough for readers to estimate that the filling degree of HOPG surface by nanoparticles is ~15-25%.

  1. From Fig 1, it can be seen that the as-synthetized nanomaterial is a little agglomerated, indicating a low and poor dispersity.

Answer: We agree with this amendment. Relevant comment about nanoparticles agglomeration has been added to the manuscript – line 115.

  1. The symbol of ÷ is wrong in the formulation like 1.2÷1.8 V, 2.2÷2.8 V, etc.

Answer: We agree with this amendment. Relevant changes have been made in the manuscript.

  1. In the nanomaterial, the CuO, Cu2O and Cu4O3 made up it. I want to know how to control their ratio and content in the nanomaterials.

Answer: The problem of tunable synthesis is out of the framework of our project. But the ratio of oxide phases in the nanomaterial – as we can conclude from the VACs analysis – is stable for synthesis conditions mentioned in the manuscript.

  1. Why did the authors say "the concentration of Cu2O and Cu4O3 strongly depend on the concentrations of all other reagents"? And what are the other reagents?

Answer: It means that in the chain of sequential reactions, pure copper and all Cu-bearing phases transform one to another. These processes can’t be considered as independent. Thus even in the case of irreversible reactions (scheme below line 219) the concentration of Cu2O will depend on the concentration of pure copper, and the concentration of Cu4O3 will depend on the concentration of Cu2O, and the concentration of CuO will depend on the concentration of Cu4O3. In the manuscript we have changed “reagents” for “Cu-based phases” to make this idea more clear – line 219.

  1. How to prove the process of Cu2O to Cu did not occur?

Answer: According to the article [39] (DOI: 10.1016/j.proci.2010.06.080) the rate constant for this reaction at room temperature is of the order E-24 s^-1, so the reaction is negligibly slow with respect to all other processes. Unfortunately the related reference was marked as [Y1] in the text of the manuscript. This mistake has been corrected – lines 260, 264, 269.

Reviewer 2 Report

The manuscript presents an interesting recent finding about the effect of electrical field on the reaction kinetics, in particular, the reduction of Cu oxide in in the CO gas environment. This research area itself still has many aspects yet to be explored, hence, the finding reported in the manuscript is highly of interest and provides a foundation for future works. 

Overall, the experimental results are missing some details to strengthen and clarify the conclusion and statements in the report. The writing of the manuscript needs major editing both in terms of the English proficiency as well as correcting the errors. Please find the comments below: 

  1. Lines 98-99: the reference for the voltage signs mentioned here may cause ambiguity. Here it is mentioned that deltaV = +1 and -1 is with respect tp the STM tip, but in line 229-230 deltaV = -1 indicates the sample negatively charged. In my perception, deltaV = -1 would mean that the STM tip is negatively charged, thus electrons flowing to the sample. Further clarification is needed. 
  2. Is there any other voltage variations used in the CO exposure experiment, in addition to +1 and -1? Such additional data will give more conclusive results.  
  3. Figure 1 only presents particle size of 1 particle. Is there any data on the particle size distribution? It will be helpful for readers.
  4. The unit of voltage in lines 123-138 is V whereas that in lines 145-162 is eV. Which one is correct?
  5. Lines 123-138 and 145-162: change the "÷" with "-" for writing 1.2-1.8 eV, for example.
  6. For data in Figure 3: how many counts are performed to obtain the histogram? Please provide some statistical information.
  7. Line 143-144: "....with the band gap up to the dimensional factor..." is not understood. Please rephrase.
  8. Line 165-167: it requires another validation to confirm the presence of metallic Cu, such as XAS, EXAFS.
  9. The Cu state distribution profiles after CO exposure in Figure 3b and 3d were obtained from sample treated differently. The former used the sample calcined in vacuum and the latter after oxidation in air. Does the results in Fig 3b can be reproduced by using the sample after oxidation in air? Please provide evidence.
  10. Any explanation for the significant increase of defect in Figure 3d?
  11. Lines 244-247, 252: What does Ref [Y1] refer to? 
  12. Lines 237-243: What is alpha and beta referring to in equations 4a and 4b?
  13. Line 243: while the equation assume small fraction of Cu-1, the experimental results shows otherwise. Expression 4 may not be valid. Please refer to the paper: J. Phys. Chem. B 2004, 108, 13667-13673 for Cu reduction mechanism in CO.
  14. Line 248: how to measure the C(CO) = 5 x 10^-8? What is the basis of this value?
  15. Lines 279-280: no evidence in the report about the effect of particle size on the CO orientation.
  16. Reference 25: the year should be 1982.

Author Response

To Reviewer #2

Many thanks for your valuable comments that helped us to improve the quality of the manuscript.

  1. Lines 98-99: the reference for the voltage signs mentioned here may cause ambiguity. Here it is mentioned that deltaV = +1 and -1 is with respect to the STM tip, but in line 229-230 deltaV = -1 indicates the sample negatively charged. In my perception, deltaV = -1 would mean that the STM tip is negatively charged, thus. Further clarification is needed.

Answer: In line 103 it is mentioned that the STM tip is grounded, so ΔV is counted with respect to the grounded tip. At deltaV= -1 electrons are flowing to the sample and the sample is negatively charged.

  1. Is there any other voltage variations used in the CO exposure experiment, in addition to +1 and -1? Such additional data will give more conclusive results.

Answer: No, we haven’t used any other voltage variations in the CO exposure experiment in this work, as we supposed this to be not very interesting. The key factor is the radius of curvature of the nanoparticles, due to which the electric field close to the surface of the nanoparticles can attain the local strength of E^9 V/m. At the same time our STM allows varying deltaV only in the range from -10 V to +10 V, so it’s impossible to change the local strength of electric field significantly by deltaV variations.   

  1. Figure 1 only presents particle size of 1 particle. Is there any data on the particle size distribution? It will be helpful for readers.

Answer: Many thanks for this amendment. We have included the data on the nanoparticle size distribution to the manuscript – Figure 1c.

  1. The unit of voltage in lines 123-138 is V whereas that in lines 145-162 is eV. Which one is correct?

Answer: In the first case we are talking about volt-ampere characteristic (VAC) of tunnel contact, so the width of zero-current section of VAC curve is measured in Volts (V). In the second case we are talking about electronic structure of the semiconductor, and the band gap is measured in Electronvolts (eV). These two values coincide up to dimensional factor. Thus if at VAC curve we observe zero-current section of 1.2 V in width (e.g.) during STS measurements of the semiconductor, one can conclude that the band gap of semiconductor under the STM tip is 1.2 eV. We have rephrased the sentence a bit to make this principle more clear – lines 152-153.

  1. Lines 123-138 and 145-162: change the "÷" with "-" for writing 1.2-1.8 eV, for example.

Answer: We agree with this amendment. Relevant changes have been added to the manuscript.

  1. For data in Figure 3: how many counts are performed to obtain the histogram? Please provide some statistical information.

Answer:  Thank you for this amendment. To obtain each histogram, at least 50 counts were performed. Relevant comment has been added to the manuscript – line 207.

  1. Line 143-144: "....with the band gap up to the dimensional factor..." is not understood. Please rephrase.

Answer: Thank you for this amendment. We have rephrased the sentence – lines 152-153.

  1. Line 165-167: it requires another validation to confirm the presence of metallic Cu, such as XAS, EXAFS.

Answer: Thank you for this amendment. Yes, XAS and EXAFS are very informative methods for catalysts diagnostics. Unfortunately the main disadvantage of these methods is the requirement of high homogeneity of the sample for unambiguous interpretation of EXAFS and XAS results. To determine the oxidation state of the elements by XAS and EXAFS, the local bonding environment of every atom should be the same in all volume of the sample. In the case of inhomogeneous distribution of various oxide phases over the surface of even single nanoparticles, the unambiguous interpretation of EXAFS and XAS results can become questionable. Our ultrahigh vacuum STM is provided with Auger electron spectrometer, and hypothetically we could use AES method to determine the changes in chemical composition of the sample surface by chemical shifts analysis. But in presence of insulators or semiconductors (e.g. surface oxides) the main reason for spectrum distortions is surface charging instead of chemical changes. Thus all these methods are inapplicable.

  1. The Cu state distribution profiles after CO exposure in Figure 3b and 3d were obtained from sample treated differently. The former used the sample calcined in vacuum and the latter after oxidation in air. Does the results in Fig 3b can be reproduced by using the sample after oxidation in air? Please provide evidence.

Answer: Chemical processes that transform system C to D (Fig. 3c and 3d) are the same as for transformation of A to B (Fig. 3a and 3b). Reduction takes place only in presence of CO molecules turned with C atom to the oxide surface. The only difference is the efficiency of the reduction process due to different concentration of CO molecules with required orientation. Thus we can conclude that system B can also be produced from system C – after oxidation of the sample in air. By the way, as we can see from Figure 3, reduced systems B and D are more similar in oxide phases ratio in comparison to initial systems A and C.

  1. Any explanation for the significant increase of defect in Figure 3d?

Answer: We suppose that the reason is not the change in concentration of the defects, but change in their observability. Maybe the interaction of oxide surface with O atom of CO molecule influence the defects in such way that Coulomb blockade takes place. Such a process can manifest itself as zero-current region at VAC curve. But this question is out of scope of our project.      

  1. Lines 244-247, 252: What does Ref [Y1] refer to?

Answer: Thanks a lot for this amendment. Ref [Y1] means ref [38]. Relevant changes have been added to the manuscript – lines 260, 264, 269.

  1. Lines 237-243: What is alpha and beta referring to in equations 4a and 4b?

Answer: Alpha and beta are parameters for the Equation 4. They depend on k1, k2, k3 and C(CO). Many thanks for this amendment. We agree that all these formulae are not very good for perception. So we changed them relevantly in order to avoid usage of r1 and r2 and to make it easy for reading – Equations 4 and 4a.

  1. Line 243: while the equation assume small fraction of Cu-1, the experimental results shows otherwise. Expression 4 may not be valid. Please refer to the paper: J. Phys. Chem. B 2004, 108, 13667-13673 for Cu reduction mechanism in CO.

Answer: Many thanks for this reference. We have included it in our paper. As for the Equation 4, having considered, we can agree with your amendment. We suppose some corrections to be necessary for our simulations. In our assumption the final stage of the reaction is irreversible, as CO2 is being pumped out of the system during the reaction. Thus the CO2 concentration and initial concentration of metal copper wouldn’t influence the reaction rate. Indeed, it would be more precise to say that Equation 4 describes not the Cred as it is, but the difference between final and initial Cred. We have made relevant corrections in our formulae (4, 6, 8, 9) and in numerical result – lines 251-259 and 285-292.

  1. Line 248: how to measure the C(CO) = 5 x 10^-8 mol/m^3? What is the basis of this value?

Answer: The pressure of CO in the vacuum chamber during the exposition was ~10^-6 mbar (line 98). Since we know the CO pressure, we can estimate the CO concentration.

  1. Lines 279-280: no evidence in the report about the effect of particle size on the CO orientation.

Answer: The influence of nanoparticle size on the orientation is indirect. From the perspective of classical electrostatics, the nanoparticle size d influences the strength of the local electric field E close to the surface of the nanoparticle, since E~1/d (lines 188-190). At the same time in our two-level approximation we have shown that E should be ~10^9 V/m for orientation effect to be observable (lines 290-291). Thus the orientation effect would be negligible for smooth metal surfaces. So the theoretical evidence for the effect of particle size on the copper reduction is that both of these estimations coincide. As for pure experimental evidence, we agree that it would be better, e.g., to investigate particles with size of 5 nm and 100 nm on the surface of one sample and to compare the reduction data. Unfortunately within the framework of our project we haven’t managed to conduct such experiment.

  1. Reference 25: the year should be 1982.

Answer: Thanks a lot for this amendment. Relevant changes have been added to the manuscript – line 413.

Round 2

Reviewer 2 Report

Thank you for providing thorough answers to the questions. The manuscript has been revised accordingly. Further checks on minor spelling errors should be conducted before the final publication.